# OpenReview forum: "Score Forgetting Distillation: A Swift, Data-Free Method for Machine Unlearning in Diffusion Models"
_ICLR.cc/2025/Conference — ICLR 2025 Poster_

### Official Review · Reviewer_hpqv · 2024-11-04

**Soundness:** 3
**Presentation:** 2
**Contribution:** 2
**Rating:** 5
**Confidence:** 3

**Summary:**

This paper presents Score Forgetting Distillation (SFD), a novel approach to machine unlearning (MU) in diffusion models (DM) that aims to "forget" undesirable information by aligning the conditional scores of "unsafe" classes with those of "safe" ones. Departing from traditional MU objectives for diffusion models, the authors introduce a score-based MU loss built upon the score distillation objective proposed in [1], which facilitates one-step synthetic data generation and obviates the need for real data. Through empirical evaluations on pretrained label-conditional and text-to-image diffusion models, the authors demonstrate the efficacy of their method in achieving targeted unlearning without compromising model quality.

[1] Score Identity Distillation: Exponentially Fast Distillation of Pretrained Diffusion Models for One-Step Generation

**Strengths:**

The exploration of machine unlearning (MU) using distilled diffusion models (DMs) is a novel approach, particularly given that distilled DMs enable significantly more efficient content generation compared to traditional DMs.

**Weaknesses:**

1. The presentation of this paper could be improved. The authors have omitted a dedicated related works section, instead integrating substantial content from prior work [1] within the methodology section. This blending of information makes it challenging to clearly identify the paper’s unique contributions.

2. The empirical study is not as comprehensive as recognized work on the same topic [2]. See questions.

[1] Score Identity Distillation: Exponentially Fast Distillation of Pretrained Diffusion Models for One-Step Generation
[2] SalUn: Empowering Machine Unlearning via Gradient-based Weight Saliency in Both Image Classification and Generation

**Questions:**

1. Regarding the first weakness, could the authors provide justification for selecting Score Identity Distillation (SiD) over Diff-Instruct and DMD in this work? Alternatively, could Diff-Instruct and DMD be integrated into the SFD framework? I encourage the authors to include related discussions or experimental results in the revised paper to address these points.

2. How were the baseline results obtained? If they were sourced from original papers, could the authors specify the exact source for these baselines? It is unclear if the setting in Table 1 aligns directly with any cases from previous work such as [1]. Please clarify, and, if applicable, identify comparable results in [1] that correspond directly with Table 1.

3. Style forgetting and random data forgetting are standard choices for MU evaluation. Why were these not considered in this study?

4. Run-time efficiency (RTE), defined as the computation time for applying an MU method, is not reported here. Additionally, the computational cost associated with pretraining a score function is not addressed. I would appreciate relevant results or discussions on these aspects.

5. Another approach to achieving similar efficiency for an unlearned model could involve performing forgetting first, followed by distillation. Could the authors comment on this alternative?

[1] SalUn: Empowering Machine Unlearning via Gradient-based Weight Saliency in Both Image Classification and Generation

---

> ### Author Response · Authors · 2024-11-19
> **Response to Reviewer hpqv (1/2)**
>
> We would like to first thank the reviewer for their detailed comments. We are glad that the reviewer finds our approach novel and valuable. Below, we address the specific concerns raised:
>
> 1. **Question:** The reviewer noted that the current presentation may obscure the true contributions of the paper and the rationale behind certain choices, such as "why SiD was selected over Diff-Instruct or DMD."
>
>     **Response:** We would like to clarify that SiD was chosen not only for *its superior stability and performance*, as demonstrated in diffusion distillation in prior works, but also for *its demonstrated compatibility with the machine unlearning (MU)* objective. Specifically:
> 	- Empirically, we observed that Diff-Instruct training is less stable when combined with the MU objective, frequently leading to mode collapse. This limitation has also been explicitly acknowledged by the authors of Diff-Instruct in their recent work, *"One-Step Diffusion Distillation through Score Implicit Matching"* [2], where they state: **"We find that DI converges fast but suffers from mode-collapse issues."** This instability makes Diff-Instruct unsuitable for our framework.
> 	- DMD, on the other hand, augments Diff-Instruct with a perceptual regression loss (LPIPS-based) to counter mode collapse and improve image quality by enforcing perceptual similarity between generated and real images. While this approach helps mitigate Diff-Instruct's instability, DMD relies on access to teacher-synthesized image data during training. This dependency adds *significant computational costs* to prepare the synthesized data required for its training.
> 	- In contrast, SiD achieves stability without relying on real or teacher-synthesized data, making it the most efficient and compatible choice for developing our framework.
>
> 2. **Question:** The reviewer raised questions regarding how baseline results, some of which were not found in their original papers, were obtained.
>
> 	**Response:** We would like to clarify that the baseline results were derived following the class-forgetting task setting described in SalUn. However, since SalUn did not release pretrained CIFAR-10/STL-10 models or their fine-tuned model weights, we reproduced their results using the official codebase, experimental configurations, and scripts provided in the paper.
>
> 	For CIFAR-10, SalUn reported the following results in Table A5 of their Appendix:
>
>     | **Methods** | **UA(↑)** | **FID(↓)** |
>     | ----------- | --------- | ---------- |
>     | Retrain     | 100.00    | 11.69      |
>     | ESD         | 100.00    | 17.37      |
>     | SalUn       | 100.00    | 11.21      |
>
> 	We reproduced these results with similar or better outcomes:
>     | **Methods** | **UA(↑)** | **FID(↓)** |
>     | ----------- | --------- | ---------- |
>     | Retrain     | 98.50     | 7.94       |
>     | ESD         | 91.21     | 12.68      |
>     | SalUn       | 99.31     | 11.25      |
>
>     The discrepancy in Unlearning Accuracy (UA) arises from differences in evaluation sample sizes. SalUn evaluated UA using only **500 samples** from the forgotten class, which makes the metric less informative since all methods achieve 100% UA under such limited conditions. To provide more precise and reliable evaluations, we increased the sample size to **5,000**. Similar adjustments were made to adapt the protocols for STL-10, ensuring consistency and fairness across all evaluations.
>
> 3. **Question:** The reviewer considered style forgetting and random data forgetting as standard choices for MU evaluation and questioned their absence in our study.
>
> 	**Response:** We believe there may be a misunderstanding here:
>
>     - Random data forgetting is *specific to image classification* tasks and classification models, such as ResNet, whereas our work focuses on image generation and diffusion models. As a result, random data forgetting is not directly relevant to this study.
> 	- Regarding style forgetting, while it is used in some MU methods for image generation, we believe our experiments (e.g., Figure 1) already effectively demonstrate the unique advantages of SFD for MU in diffusion models, making additional style forgetting evaluations unnecessary for the scope of this paper.
>
> [2] Luo, Weijian, et al. "One-step diffusion distillation through score implicit matching." *arXiv preprint arXiv:2410.16794* (2024).

---

> > ### Author Response · Authors · 2024-11-19
> > **Response to Reviewer hpqv (2/2)**
> >
> > 4. **Question:** The reviewer expressed interest in understanding the runtime efficiency and computational cost associated with pretraining a score function.
> >
> > 	**Response:** We clarify that the computational cost of pretraining a score function is consistent across all finetuning-based MU methods, as they all rely on the same pretrained diffusion model. By Tweedie’s formula, a pretrained diffusion model can be directly converted into a pretrained score function, making the two interchangeable. For CIFAR-10, the DDPM pretraining required **800K training steps**, while for STL-10, it required **250K steps**. In the case of Stable Diffusion, exact pretraining details are not publicly available, but its Hugging Face page specifies that it required approximately **200,000 GPU hours** on $\underline{\text{NVIDIA A100 PCIe 40GB GPUs}}$. Once a pretrained diffusion model (i.e., score function) is available, SFD fine-tuning required only **96–128 GPU hours** on $\underline{\text{NVIDIA RTX A5000 GPUs}}$, highlighting its computational efficiency compared to the cost of pretraining.
> >
> > 5. **Question:** The reviewer expressed interest in our opinion on an alternative approach of "forgetting first, then distillation."
> >
> > 	**Response:** While this approach is plausible, we believe it is inherently suboptimal. Distilling an unlearned model obtained from baseline MU methods would carry forward the limitations of the initial MU method, which often acts as a bottleneck. In contrast, SFD integrates distillation directly into the unlearning process, effectively bypassing this bottleneck. Our experimental results demonstrate that SFD consistently outperforms baseline methods in both unlearning accuracy and generation quality, suggesting that a "forgetting first, then distillation" approach is unlikely to achieve comparable performance.

---

> ### Author Response · Authors · 2024-11-22
> **Follow-up on Discussion**
>
> Dear Reviewer hpqv,
>
> We sincerely thank you again for your detailed comments and valuable feedback. In our responses, we have provided clarifications on:
>
> - the rationale for selecting our score distillation method;
> - the sources of baseline results;
> - the specific choices of MU experiments;
> - the run-time efficiency and computational cost of pretraining;
> - the comparison to the "forget first, then distill" approach.
>
> With the discussion period nearing its end, we would greatly appreciate any remaining thoughts or questions you might have. If our responses have satisfactorily addressed your concerns, we kindly hope you will reconsider your evaluation of our work in light of the clarifications provided.
>
> Warm regards,
>
> Authors of Submission #5206

---

> > ### Comment · Reviewer_hpqv · 2024-11-25
> >
> > Dear authors,
> >
> > Thank you for your detailed response. After carefully reviewing your reply and considering comments from other reviewers, I find that some of my concerns have been addressed, while others remain unresolved.
> >
> > Regarding Q1, while the authors have provided justifications for the advantages of SiD over other one-step diffusion models (DMs), I strongly encourage the authors to empirically evaluate alternative one-step DMs within the SFD framework. Since the main contribution of this work lies in the SFD framework rather than SiD itself, such evaluations using SFD with other popular one-step DMs would make the study more comprehensive and robust.
> >
> > I am overall satisfied with the authors' responses to Q2-Q5 and encourage the authors to revise the manuscript accordingly based on these clarifications.
> >
> > In summary, I will retain my original rating and confidence level for now. The primary reason for not increasing my rating is my remaining concern regarding Q1, a concern shared by another reviewer. Additionally, I have not increased my confidence level, as I am not deeply familiar with one-step DMs and cannot make a definitive judgment without empirical evaluations specific to the exact MU scenario.

---

> ### Author Response · Authors · 2024-11-26
>
> Dear Reviewer hpqv,
>
> Thank you for your thoughtful feedback and engagement in the discussion. We are glad to hear that our responses to Q2–Q5 addressed your concerns.
>
> Regarding Q1, we appreciate your suggestion to evaluate the proposed SFD framework using alternative diffusion distillation methods. Our initial focus was on integrating the model-based Fisher divergence into the SFD framework, driven by compelling evidence of its superior performance (e.g., SiD) over KL-divergence-based distillation methods (e.g., Diff-Instruct and DMD), as well as its ability to deliver strong results with the proposed modifications, effectively outperforming existing approaches. We agree that evaluating SFD with alternative methods could further demonstrate the robustness and versatility of our approach.
>
> In response to your suggestion, we conducted a new experiment on CIFAR-10, implementing a variant of our framework called **SFD-KL**. This version modifies *the model-based Fisher divergence-based loss and SiD-based gradient* in SFD to use *a KL-based loss and Diff-Instruct-based gradient*. Below are the evaluation results:
>
> | **Model**  | **UA (↑)** | **FID (↓)** | **IS (↑)** | **Precision (↑)** | **Recall (↑)** | **NFEs (↓)** | **Data-free** |
> | ---------- | ---------- | ----------- | ---------- | ----------------- | -------------- | ------------ | ------------- |
> | **SFD**    | **99.64**  | **5.35**    | **9.51**   | 0.6587            | **0.5471**     | **1**        | **Y**         |
> | **SFD-KL** | 99.53      | 6.99        | 9.44       | **0.6688**        | 0.5016         | **1**        | **Y**         |
>
> Additional details are provided in the newly revised $\underline{\text{Appendix B.8}}$. These results highlight the flexibility of the **SFD framework, demonstrating its ability to integrate alternative score distillation methods while maintaining strong performance with unlearned, one-step distilled diffusion models**. Overall, the original SFD achieves superior results, reinforcing the rationale behind our original design choices.
>
> We hope this new evaluation enhances the comprehensiveness and robustness of our study. If you have any further questions or suggestions for additional experiments, please don’t hesitate to let us know. We are fully committed to addressing any concerns from you and other reviewers throughout the discussion period.
>
> Warm regards,
>
> Authors of Submission #5206

---

> > ### Comment · Reviewer_hpqv · 2024-11-27
> >
> > Thanks for your further clarification.

---

> > > ### Author Response · Authors · 2024-11-27
> > >
> > > If our clarifications have successfully addressed your questions, we would greatly appreciate it if you could consider raising your score. Thank you!

---

> ### Author Response · Authors · 2024-12-01
> **Official Comment by Authors**
>
> Dear Reviewer hpqv,
>
> If our response and revised manuscript have effectively addressed your concerns, we kindly request that you reevaluate our work based on the updated information. We would greatly appreciate your consideration in potentially raising your score.
>
> Thank you for your time and thoughtful review.
>
> Sincerely,
>
> The Authors

---

### Official Review · Reviewer_5B7n · 2024-11-04

**Soundness:** 3
**Presentation:** 4
**Contribution:** 2
**Rating:** 5
**Confidence:** 5

**Summary:**

The paper introduces Score Forgetting Distillation (SFD), a data-free method for machine unlearning (MU) in diffusion models. The goal is to efficiently erase specific classes or concepts (e.g., "unsafe" or undesirable content) from pre-trained diffusion models without accessing the original training data. SFD achieves this by aligning the conditional scores of the target classes or concepts to forget with those of safe or desired ones, effectively overriding the unwanted information. The method incorporates a score-based MU loss into the score distillation process, which not only enables data-free unlearning but also accelerates the generation speed of diffusion models by distilling them into one-step generators. The authors validate the effectiveness of SFD through experiments on class-conditional diffusion models (e.g., CIFAR-10 and STL-10) and text-to-image models like Stable Diffusion, demonstrating that their approach can effectively forget specific classes or concepts while preserving the quality and diversity of the remaining content.

**Strengths:**

1. The paper presents a novel approach to machine unlearning in diffusion models by introducing Score Forgetting Distillation. Unlike previous methods that require access to real data or rely on stringent assumptions, SFD operates in a data-free manner and integrates unlearning directly into the distillation process. This creative combination of score distillation and machine unlearning extends to both class-conditional and text-to-image diffusion models.

2. The proposed method is thoroughly evaluated on multiple datasets and models, including CIFAR-10, STL-10, and Stable Diffusion. The experiments include both quantitative metrics (e.g., Unlearning Accuracy, FID, IS, Precision, Recall) and qualitative results (e.g., visual examples of forgetting effects). Ablation studies analyze the impact of different components of the method, enhancing the robustness of the evaluation.

3. The paper is well-written and organized, with clear explanations of the problem, methodology, and experimental results. Figures and tables are informative and effectively support the text. The inclusion of algorithm pseudo-code and detailed descriptions aids in understanding the implementation details.

4. Machine unlearning in diffusion models is crucial for enhancing the safety and trustworthiness of generative AI. By providing a data-free method that accelerates both unlearning and sampling, the paper contributes to making diffusion models more practical and secure for real-world applications.

**Weaknesses:**

1. While the paper introduces a novel method, the contributions may be perceived as incremental within the context of existing work on score distillation [1] and machine unlearning [2]. The proposed method builds upon recent advancements in data-free diffusion distillation, with the primary novelty being the integration of machine unlearning into this framework. Providing additional emphasis on the theoretical foundations or discussing broader implications could strengthen the overall contribution.

2. Although SFD reduces the number of sampling steps, this benefit appears to be a general characteristic of score distillation or matching methods. The authors do not sufficiently explain the specific advantages of integrating unlearning into score distillation. A more intuitive and necessary ablation study would involve comparing the results with the following baselines: (1) comparing SFD against existing unlearning methods applied to one-step diffusion models (e.g., SiD/DMD distilled models [1,3,4] or pre-trained consistency models [5,6]); (2) applying score distillation or matching methods to models that have already undergone unlearning. Additionally, the authors could compare their method with baseline unlearned models using fewer sampling steps (e.g., 50-step DDIM [7] or 10-step DPM-Solver++ [8,9]). Intuitively, reducing the number of sampling steps should primarily affect image quality rather than unlearning accuracy.

3. Machine unlearning generally focuses on methods for rapid modification of existing models. An essential aspect of evaluations is the algorithm's efficiency [10]. If the training time of the proposed algorithm is excessively long, one might question why not simply filter out all harmful data and retrain the model from scratch. However, the paper does not provide an analysis of the algorithm's runtime or computational efficiency. If this is a misunderstanding on my part, clarification would be appreciated.

4. A major application of machine unlearning is its direct deployment in existing products for hot updates to production models, which requires the algorithm to handle sequential unlearning requests effectively [11]. However, the SFD method presented by the authors has only been applied to models that have not undergone prior distillation. Does SFD perform equally well when applied to distilled models? For example, does it work effectively when sequentially forgetting classes such as airplane, bird, car, cat, and deer from the STL-10 dataset? In such scenarios, does it perform better than other unlearning methods?

5. There appear to be inconsistencies in the experimental results. For instance, in Table 1, the authors compare their method with ESD, SA, and SalUn, whereas in Tables 2 and 3, some of these baselines are missing. Could the authors briefly explain the reason for excluding them?

> [1] Zhou, Mingyuan, et al. "Score identity distillation: Exponentially fast distillation of pretrained diffusion models for one-step generation." Forty-first International Conference on Machine Learning. 2024.
>
> [2] Kumari, Nupur, et al. "Ablating concepts in text-to-image diffusion models." Proceedings of the IEEE/CVF International Conference on Computer Vision. 2023.
>
> [3] Yin, Tianwei, et al. "One-step diffusion with distribution matching distillation." Proceedings of the IEEE/CVF Conference on Computer Vision and Pattern Recognition. 2024.
>
> [4] Yin, Tianwei, et al. "Improved Distribution Matching Distillation for Fast Image Synthesis." arXiv preprint arXiv:2405.14867 (2024).
>
> [5] Song, Yang, et al. "Consistency models." Proceedings of the 40th International Conference on Machine Learning. 2023.
>
> [6] Song, Yang, and Prafulla Dhariwal. "Improved Techniques for Training Consistency Models." The Twelfth International Conference on Learning Representations.
>
> [7] Song, Jiaming, Chenlin Meng, and Stefano Ermon. "Denoising Diffusion Implicit Models." International Conference on Learning Representations.
>
> [8] Lu, Cheng, et al. "Dpm-solver: A fast ode solver for diffusion probabilistic model sampling in around 10 steps." Advances in Neural Information Processing Systems 35 (2022): 5775-5787.
>
> [9] Lu, Cheng, et al. "Dpm-solver++: Fast solver for guided sampling of diffusion probabilistic models." arXiv preprint arXiv:2211.01095 (2022).
>
> [10] Fan, Chongyu, et al. "SalUn: Empowering Machine Unlearning via Gradient-based Weight Saliency in Both Image Classification and Generation." The Twelfth International Conference on Learning Representations.
>
> [11] Zhang, Yihua, et al. "Unlearncanvas: A stylized image dataset to benchmark machine unlearning for diffusion models." arXiv preprint arXiv:2402.11846 (2024).

**Questions:**

See weaknesses.

---

> ### Author Response · Authors · 2024-11-19
> **Response to Reviewer 5B7n (1/2)**
>
> We would like to thank the reviewer for their detailed and insightful comments on our work. We are glad that the reviewer regards SFD as *a novel and creative approach* backed by thorough evaluation and appreciates *the practical contributions* of our work to real-world applications. Below, we address the concerns raised:
>
> 1. **Weakness:** The contributions of the proposed method may be perceived as incremental, as it builds upon existing advancements in score distillation and machine unlearning.
>
>     **Response:** Regarding the level of perceived contributions, we strongly emphasize that our work establishes a brand-new framework for machine unlearning (MU), introducing an efficient and effective solution to the long-standing challenge of data-free MU. This contribution is far from incremental. To the best of our knowledge, no prior work has sufficiently addressed the problem of data-free MU for diffusion models, a task rendered especially challenging due to the inefficiency of multistep sampling in traditional approaches. By leveraging score distillation for MU tasks, our method not only overcomes this barrier but also enables a practical solution that opens new avenues for future research. Beyond the immediate application, our framework provides a foundation for adapting unlearning techniques to other distilled diffusion models, significantly advancing the field.
>
> 2. **Weakness:** The authors do not sufficiently explain the specific advantages of integrating unlearning into score distillation and fail to compare with other one-step diffusion models or alternative multi-step baselines (e.g., SiD/DMD or diffusion solvers like DDIM).
>
> 	**Response:** We clarify the following:
>
> 	**i.** To the best of our knowledge, no existing MU methods, apart from our own, have been or can be directly applied to the distilled diffusion models mentioned by the reviewer. While we acknowledge the possibility that such methods could exist, demonstrating their applicability lies beyond the scope and burden of this paper. Specifically, existing finetuning-based MU methods are tailored for standard multistep diffusion models and rely on modifying the diffusion loss (e.g., MSE) to erase "unsafe" classes or concepts. In contrast, distilled diffusion models are not trained using these loss formulations, making such methods inherently incompatible.
>
> 	**ii.** Distilling an unlearned diffusion model is possible but inherently limited by the performance of the teacher model. For example, a baseline MU method applied to a standard diffusion model would produce a teacher model that our method already outperforms in terms of unlearning accuracy and generation quality. While distillation alone can accelerate sampling speed, it cannot address the fundamental limitations inherent in the teacher model.
>
> 	**iii.** Using diffusion solvers with fewer steps, such as DDIM or DPM-Solver++, can reduce sampling time but does not approach the efficiency of our one-step SFD. These solvers also come at the cost of degraded sample quality, as noted by the reviewer. In contrast, SFD achieves significant advantages across sampling speed, unlearning accuracy, and generation quality, demonstrating a comprehensive superiority over alternative approaches.
>
> 3. **Weakness:** The paper does not provide an analysis of the algorithm's runtime or computational efficiency.
>
>     **Response:** We clarify that SFD finetuning is highly efficient, requiring far fewer steps than pretraining. For instance, pretraining the CIFAR-10 DDPM model required **800K steps**, while SFD finetuning required only **50K steps**. Furthermore, our two-stage approach, SFD-Two Stage, achieves competitive performance in just **1.5K steps**. For Stable Diffusion models, SFD required only **25K steps** (celebrity forgetting) and **37K steps** (nudity forgetting) with a batch size of 8. These results demonstrate the efficiency of our approach compared to retraining from scratch, making SFD practical for real-world applications.

---

> > ### Author Response · Authors · 2024-11-19
> > **Response to Reviewer 5B7n (2/2)**
> >
> > 4. **Weakness:** The reviewer expressed concerns that SFD might be limited to models that have not undergone prior distillation and may therefore struggle with sequential unlearning.
> >
> >     **Response:**  We would like to address this concern by clarifying that SFD has already been successfully applied to distilled models, as detailed in our **"SFD-Two Stage"** approach in $\underline{\text{Section 3}}$. While we acknowledge the importance of sequential unlearning, we are not aware of it being a standard evaluation protocol adopted by existing baselines. Although this specific application scenario lies beyond the current scope of our paper, the demonstrated ability of SFD to safeguard pre-distilled models already provides strong evidence of its potential applicability to sequential settings. We are open to exploring this further in our future work and, if the reviewer deems this crucial for improving their evaluation, we will make our best effort in the next few days to include additional experiments demonstrating how SFD performs in sequential unlearning scenarios.
> >
> > 5. **Weakness:** The reviewer raised concerns regarding the absence of certain baseline results in specific tables.
> >
> >     **Response:** We would like to clarify that both ESD and SA results are included in Tables 1 and 2. However, SalUn appears in Table 1 but not in Table 2 because SalUn did not conduct experiments on the celebrity forgetting task, unlike the other baselines. We strive to maintain consistency wherever possible and will explicitly clarify this in the revised manuscript to avoid any confusion.

---

> ### Author Response · Authors · 2024-11-22
> **Follow-up on Discussion**
>
> Dear Reviewer 5B7n,
>
> We sincerely thank you again for your valuable suggestions and detailed feedback. To address your concerns, we have provided clarifications on:
>
> - the significance of our method's contributions;
> - the distinctive advantages of our approach;
> - the efficiency of our fine-tuning process;
> - the primary focus of our approach and its potential for other tasks;
> - the consistency of our experimental results.
>
> With the discussion period nearing its end, we would greatly appreciate any remaining thoughts or questions you might have. If our responses have addressed your concerns, we kindly hope you will reconsider your evaluation of our work in light of the clarifications provided.
>
> Warm regards,
>
> Authors of Submission #5206

---

> > ### Author Response · Authors · 2024-11-27
> >
> > Dear Reviewer 5B7n,
> >
> > We eagerly await your feedback and would greatly appreciate it if you could acknowledge reading our responses and reevaluate our paper based on them and the provided revisions.
> >
> > Thank you,
> >
> > The Authors of Score Forgetting Distillation

---

> ### Author Response · Authors · 2024-12-01
> **Official Comment by Authors**
>
> Dear Reviewer 5B7n,
>
> We hope this message finds you well. While we have not yet received your feedback during the rebuttal, we sincerely hope you have had a chance to review our responses and the provided revisions. We are eager to use the discussion period to address any remaining or new concerns you may have and incorporate them into our revisions.
>
> If our responses and the revised manuscript have adequately addressed your questions, we kindly request that you consider reevaluating our work based on the updated information. Thank you for your time and thoughtful consideration.
>
> Sincerely,
>
> The Authors

---

### Official Review · Reviewer_KevT · 2024-11-05

**Soundness:** 2
**Presentation:** 2
**Contribution:** 2
**Rating:** 8
**Confidence:** 5

**Summary:**

This paper introduces Score Forgetting Distillation (SFD), a method that enables diffusion models to unlearn unsafe content by aligning it with safe content, without needing real data. SFD improves generation speed and maintains quality for other concepts, supporting safer generative AI.

**Strengths:**

1. Covers multiple diffusion model unlearning tasks, such as object classes, celebrity identities, and nudity.
2. Has a strong theoretical foundation.

**Weaknesses:**

1. There is no related work section to highlight the differences among existing works.
2. Evaluation under adversarial prompt attacks, such as UnlearnDiffAtk [1], is missing, even though it is a common evaluation tool for unlearned diffusion models.
3. A comparison of generation times is missing, which is essential to demonstrate the efficiency of the proposed one-step generator.
4. The baselines used in the experiments are weak, and stronger state-of-the-art methods, such as AdvUnlearn [2], should be considered.

[1]  To Generate or Not? Safety-Driven Unlearned Diffusion Models Are Still Easy To Generate Unsafe Images … For Now, ECCV’24
[2]  Defensive Unlearning with Adversarial Training for Robust Concept Erasure in Diffusion Models, NeurIPS’24

**Questions:**

check Weakness section

---

> ### Author Response · Authors · 2024-11-19
> **Response to Reviewer KevT (1/2)**
>
> We sincerely thank the reviewer for their thoughtful feedback and for highlighting important aspects to strengthen our work. Below, we provide detailed responses, emphasizing the unique strengths of our approach while addressing potential misunderstandings.
>
> 1. **Weakness:** There is no related work section to highlight the differences among existing works.
>
>     **Response:** Thank you for suggesting a related work section to highlight our method's novelty and differences among existing works. We have included a dedicated related work section in $\underline{\text{Appendix A}}$ to discuss these differences in the revised manuscript.
>
> 2. **Weakness:** Evaluation under adversarial prompt attacks, such as UnlearnDiffAtk [1], is missing, even though it is a common evaluation tool for unlearned diffusion models.
>
>     **Response:** Thank you for bringing up the importance of adversarial prompt attack evaluations. While SFD was not explicitly designed to counter such attacks, **it demonstrates *strong inherent robustness* against UnlearnDiffAtk [1], significantly outperforming all other unlearned UNet-based baselines in terms of attack success rate (ASR)**. This highlights the robustness benefits of our UNet finetuning approach, even without specialized adversarial defenses.
>
>     It is important to note that adversarial prompt attacks like UnlearnDiffAtk primarily exploit vulnerabilities in the text encoder. As SFD focuses on finetuning the UNet, the unchanged text encoder remains susceptible to these attacks. However, this does not undermine the contributions of our method: SFD provides a *data-free, efficient, and robust unlearning solution* with a one-step sampling process that delivers unmatched generation speed compared to multi-step baselines. Furthermore, **our method is *orthogonal* to approaches like AdvUnlearn [2], which focus on finetuning the text encoder**. This orthogonality opens the door for complementary integration. Specifically, we can directly combine AdvUnlearn’s defended text encoder with SFD’s UNet finetuning to further enhance robustness, achieving state-of-the-art performance. We refer to the version of SFD with its text encoder replaced by that of AdvUnlearn-TE as **SFD-TE**. Below is a comparison of the attack success rate (ASR↓), where lower values indicate better robustness:
>     | Scenario           |   ESD   |   FMN   |   SLD   | AdvUnlearn-UN | AdvUnlearn-TE | SFD (ours) | SFD+TE (ours) |
>     |:---------------------|:-------:|:-------:|:-------:|:-------------:|:-------------:|:----------:|:-------------:|
>     | No attack          | 20.42%  | 88.03%  | 33.10%  | -             | 7.75%            | 7.04%      | **0.70%**     |
>     | UnlearnDiffAtk [1] | 76.05%  | 97.89%  | 82.39%  | 64.79%        | 21.13%        | 55.63%     | **7.04%**     |
>
>     **Key observations:**
>
>     - **SFD** achieves the best robustness among all unlearned UNet models under adversarial conditions.
>     - **SFD+TE** further improves robustness, reducing ASR to **0.70%** in the absence of attacks and maintaining strong performance with an ASR of **7.04%** under UnlearnDiffAtk.
>
>     These results validate SFD’s robustness as a standalone method while demonstrating its potential for seamless integration with complementary approaches like AdvUnlearn. Crucially, this integration does not compromise SFD’s simplicity or its significant efficiency advantage over multi-step methods.
>
> 3. **Weakness:** A comparison of generation times is missing, which is essential to demonstrate the efficiency of the proposed one-step generator.
>
>     **Response:** Thank you for raising this point. SFD’s one-step generator offers **unmatched efficiency** compared to multi-step baselines. Specifically, for text-to-image generation based on Stable Diffusion, SFD generates an image in approximately **0.14 seconds** per prompt, whereas finetuning-based multi-step models require approximately **~5.46 seconds** per image per prompt using a 50-step DDIM sampler and classifier-free guidance with a scale of 7.5. All benchmarks were conducted on a single $\underline{\text{24GB NVIDIA RTX A5000 GPU}}$. This dramatic reduction in generation time underscores the practicality of SFD for real-world applications, combining high efficiency with competitive robustness and quality.

---

> > ### Author Response · Authors · 2024-11-19
> > **Response to Reviewer KevT (2/2)**
> >
> > 4. **Weakness:** The baselines used in the experiments are weak, and stronger state-of-the-art methods, such as AdvUnlearn [2], should be considered.
> >
> >    **Response:** We respectfully disagree with the assertion that our baselines are weak. Although we acknowledge that there exist more recent methods with certain advantages in terms of empirical adversarial robustness against certain type of attacks (*e.g.,* adversarial prompts), such as AdvUnlearn [2], methods like AdvUnlearn have a different focus on the adversarial robustness side. For example, AdvUnlearn adopts an adversarial training framework, which combines ESD [3] for concept erasure and UnlearnDiffAtk [4] for adversarial prompt generation. It aims to enhance the model's robustness against certain types of attacks (i.e., UnlearnDiffAtk) without sacrificing much of the model's image generation quality. In terms of FID (one key metric measuring generation quality), AdvUnlearn already underperforms ESD, which is one of our baselines. **Additionally, AdvUnlearn focuses on defending the text encoder, whereas SFD targets the SD-UNet, making it an *orthogonal approach* to our method.** Based on the Table in our Response A2, our SFD model can achieve best adversarial robustness among UNet baselines, and the SFD+TE variant can further achieve state-of-the-art performance among all the baselines including AdvUnlearn, demonstrating the flexibility and robustness of SFD.
> >
> > [3] Rohit Gandikota, Joanna Materzynska, Jaden Fiotto-Kaufman, and David Bau. Erasing Concepts from Diffusion Models. arXiv preprint arXiv:2303.07345, 2023.

---

> ### Author Response · Authors · 2024-11-22
> **Follow-up on Discussion**
>
> Dear Reviewer KevT,
>
> We sincerely thank you again for your valuable feedback and suggestions. In response, we have conducted additional experiments, revised our manuscript, and included new results to address the points you raised. With the discussion period nearing its end, we would greatly appreciate any additional thoughts or questions you might have. We hope that our efforts have clarified the key aspects of our work and positively informed your ongoing evaluation.
>
> Warm regards,
>
> Authors of Submission #5206

---

> > ### Comment · Reviewer_KevT · 2024-11-24
> >
> > Thank you for providing detailed experimental results and explanations. All my concerns have been addressed, so I am raising my rating to 8.  Please ensure the newly conducted experimental results and insights are incorporated into the revision.

---

> > > ### Author Response · Authors · 2024-11-24
> > >
> > > Thank you for your positive feedback and for raising your rating. We’re delighted to hear that our responses have fully addressed your concerns. We will ensure that the newly conducted experimental results and insights are thoroughly incorporated into the revised manuscript.

---

### Official Review · Reviewer_4Bv9 · 2024-11-07

**Soundness:** 3
**Presentation:** 3
**Contribution:** 3
**Rating:** 8
**Confidence:** 4

**Summary:**

The paper introduces Score Forgetting Distillation (SFD), a novel, data-free method for machine unlearning in diffusion models. SFD enables the efficient removal of specific classes or concepts from pretrained diffusion models without requiring access to real training data. It works by aligning the conditional scores of the undesired ("unsafe") content with those of desired ("safe") content within the model's score function.

**Strengths:**

Several strengths:

1. Novelty of the Approach: The paper introduces a novel method for machine unlearning in diffusion models using score distillation. To the best of current knowledge, this is the first work that approaches class and concept unlearning from the score distillation perspective. This innovative method opens up new avenues for efficient and effective unlearning in generative models.

2. Clarity of Presentation: The authors present their ideas and methods clearly, which helps readers understand both the existing literature and the proposed approach. The paper effectively explains complex concepts related to diffusion models and machine unlearning, making it accessible to a broader audience.

3. Extensive Experiments: The paper includes comprehensive experiments across various unlearning scenarios. These experiments not only demonstrate the effectiveness of the proposed method but also provide valuable insights into its applicability in different contexts. The thorough empirical evaluation strengthens the credibility of the findings

**Weaknesses:**

Weakness:

Running Time Concerns: There is a concern regarding the overall runtime of the unlearning process proposed in the paper. While the method aims to accelerate forgetting and improve generation speed, it is important to evaluate the computational efficiency and practicality of the unlearning procedure. Without empirical data on the time required for the process, it's unclear whether the method is suitable for large-scale or real-time applications.

Lack of Robustness Evaluation Against Adversarial Prompts: Previous studies [1,2] have shown that adversarial text prompts can circumvent unlearning mechanisms in text-to-image (T2I) models. The paper lacks an evaluation of how robust the proposed method is against such adversarial inputs. Assessing the method's ability to prevent the generation of unlearned content when faced with intentionally manipulative prompts is crucial. Providing robustness evaluations or proposing solutions to address potential vulnerabilities would strengthen the validity and reliability of the unlearning approach.

1. Zhang Y, Jia J, Chen X, et al. To generate or not? safety-driven unlearned diffusion models are still easy to generate unsafe images... for now[C]//European Conference on Computer Vision.

2.Pham M, Marshall K O, Cohen N, et al. Circumventing concept erasure methods for text-to-image generative models[C]//The Twelfth International Conference on Learning Representations. 2023.

**Questions:**

Is it possible to provide the running time for the whole unlearning procedure compared with others?

Is it possible to provide some robustness evaluation on T2I unlearned models?

---

> ### Author Response · Authors · 2024-11-19
> **Response to Reviewer 4Bv9**
>
> We sincerely thank the reviewer for their detailed and insightful comments on our work. We are delighted that the reviewer recognizes *the novelty of our pioneering approach* to machine unlearning from the perspective of score distillation, as well as *the clarity of our presentation* and *the comprehensiveness of our experiments*.
>
> Below, we present detailed clarifications and new evidence to address the two key weaknesses and questions raised by the reviewer. These results, which will be incorporated into our revision, further confirm that our unlearning procedure is not only fast—particularly when leveraging a pre-distilled generator—but also complements existing unlearning methods applied to the text encoder and excels in the recommended robustness evaluations, demonstrating its effectiveness and practicality.
>
> 1. **Weakness:** The reviewer raised a valid concern regarding the lack of detailed explanations on runtime, especially given our ambitious goal of simultaneously accelerating diffusion models and unlearning "unsafe" classes or concepts effectively.
>
>     **Response:**  We would like to emphasize that SFD fine-tuning incurs a negligible runtime compared to pretraining. For instance, training a CIFAR-10 class-conditional diffusion model required **800K steps**, whereas SFD fine-tuning needed only **50K steps**. Furthermore, we propose a two-stage approach (i.e., first distill, then forget), referred to as **"SFD-Two Stage"** in Section 3 of the paper, which offers even greater efficiency when a pre-distilled generator is available. Starting with a pre-distilled diffusion model (via SiD), SFD fine-tuning can further achieve over a **10× acceleration**, requiring only **1.5K steps** to attain strong performance. Specifically, as demonstrated in $\underline{\text{Figure 7 of Appendix B.3}}$, SFD-Two Stage on CIFAR-10 achieves a Unlearning Accuracy (UA) of **99.5%** and a Fréchet Inception Distance (FID) of **5.73** with as few as 1.5K steps. These results underscore the practicality and computational efficiency of our method for large-scale unlearning tasks.
>
> 2. **Weakness:** The reviewer highlighted the lack of robustness evaluation against adversarial prompts as a key weakness of our paper.
>
>     **Response:** We deeply appreciate this suggestion and agree that such evaluations can further strengthen the case for the effectiveness of SFD. Adversarial prompts indeed present a significant challenge to unlearned models. However, due to the nature of our method—distilling a multistep denoising diffusion model into a one-step generator—certain adversarial attacks proposed in prior studies, such as circumventing concept erasure [2], may not directly apply to our framework. Nonetheless, we have conducted additional experiments using UnlearnDiffAtk [1] to evaluate the robustness of our method. Below, we present a comparison of the *attack success rate* (**ASR**), where lower values indicate better robustness.
>
>     | Scenario           |   ESD   |   FMN   |   SLD   | AdvUnlearn-UN | AdvUnlearn-TE | SFD (ours) | SFD+TE (ours) |
>     |:---------------------|:-------:|:-------:|:-------:|:-------------:|:-------------:|:----------:|:-------------:|
>     | No attack          | 20.42%  | 88.03%  | 33.10%  | -             |  7.75%             | 7.04%      | **0.70%**     |
>     | UnlearnDiffAtk [1] | 76.05%  | 97.89%  | 82.39%  | 64.79%        | 21.13%        | 55.63%     | **7.04%**     |
>
>     Importantly, we also emphasize that complementary approaches, such as adversarial training on the text encoder, can be seamlessly integrated with SFD to further enhance its robustness. Specifically, we introduce SFD+TE, which replaces the frozen text encoder in SFD with the adversarially trained text encoder from AdvUnlearn-TE [3]. In this table, AdvUnlearn [3] refers to an adversarial training-based method for machine unlearning, with "AdvUnlearn-UN" and "AdvUnlearn-TE" representing its application to the Stable Diffusion UNet and text encoder, respectively. **While AdvUnlearn-UN and SFD both target the UNet, making them not directly complementary, AdvUnlearn-TE addresses the text encoder, which is *orthogonal* to our UNet-centric approach.** The introduced SFD+TE achieves exceptional robustness, reducing the ASR to **0.70%** in the absence of attacks and maintaining strong performance with an ASR of **7.04%** under adversarial conditions. These results highlight the complementary nature of adversarial training-based methods and SFD, showcasing how these approaches together can tackle robustness challenges in unlearning more effectively.
>
> [3] Zhang, Yimeng, et al. "Defensive Unlearning with Adversarial Training for Robust Concept Erasure in Diffusion Models." *arXiv preprint arXiv:2405.15234* (2024).

---

> ### Author Response · Authors · 2024-11-22
> **Follow-up on Discussion**
>
> Dear Reviewer 4Bv9,
>
> We sincerely thank you again for your valuable feedback and suggestions. In response, we have provided clarifications, conducted additional experiments, and included new results to address the points you raised. With the discussion period nearing its end, we would greatly appreciate any additional thoughts or questions you might have. We hope that our efforts have clarified the key aspects of our work and positively informed your ongoing evaluation.
>
> Warm regards,
>
> Authors of Submission #5206

---

> > ### Comment · Reviewer_4Bv9 · 2024-11-27
> >
> > Thank you for providing detailed experimental results and explanations. You have addressed all my concerns, so I am raising my rating to 7. Please ensure the new experimental results and insights are included in the revision.

---

> > > ### Author Response · Authors · 2024-11-27
> > >
> > > We sincerely appreciate your acknowledgment of our responses, additional results, and explanations through your improved rating. We remain committed to further refining the paper by integrating all the new results and valuable insights gained from our discussions with the reviewers.

---

### Comment · Area_Chair_EHHw · 2024-11-25
**The author-reviewer discussion period is ending soon**

Dear reviewers,

If you haven’t done so already, please engage in the discussion as soon as possible. Specifically, please acknowledge that you have thoroughly reviewed the authors' rebuttal and indicate whether your concerns have been adequately addressed. Your input during this critical phase is essential—not only for the authors but also for your fellow reviewers and the Area Chair—to ensure a fair evaluation.
Best wishes,
AC

---

### Meta-Review · Area_Chair_EHHw · 2024-12-21

**Metareview:**

This paper received mixed reviews, with both positive and negative reviewers maintaining their positions throughout the discussion. The positive reviewers praised the quality, significance, and empirical results of the work. In contrast, the negative reviewers raised concerns about the novelty of the method, the computational cost due to heavy training time, and the unclear motivation for performing distillation and unlearning simultaneously (led to the request for additional comparisons with baselines that first perform unlearning and then apply one-step distillation).

After carefully considering the discussion, I find myself more aligned with the positive reviewers for the following reasons:

Novelty and Technical Contribution: While the proposed method may appear to be an incremental combination of selective distillation (SiD) and an unlearning method, implementing it in practice requires substantial effort. For instance, framing the process of simultaneous distillation and unlearning as a bi-level optimization problem is non-trivial and involves careful design and relaxation. This level of technical depth adds value beyond a simple combination of existing techniques.

Motivation for Simultaneous Distillation and Unlearning: While the authors have not fully clarified the motivation behind combining these two objectives, I believe there is merit; the literature suggests that safety-aligned pre-trained models may lose their safety guarantees during subsequent fine-tuning [1,2]. This highlights a potential advantage of simultaneous distillation and unlearning, as it could prevent unexpected behavior when we try to first unlearn a diffusion model and then distill later. I encourage the authors to better articulate and empirically validate this motivation in their future work.

Training Time Concerns: The reviewers raised concerns about the computational cost of the proposed method. I think the authors' response to this is reasonable. Provided that the cost of the proposed method is reasonable compared to the sum of costs for the unlearning and distillation.

Overall, despite some open questions and areas where the paper could provide more clarity, the contributions of this work are meaningful, and the method demonstrates potential for advancing the field. Therefore, I recommend acceptance, while strongly encouraging the authors to explicitly clarify the motivation for simultaneous distillation and unlearning.

[1] Qi et al., Fine-tuning aligned language models compromises safety, even when users do not intend to!, 2023.
[2] Kim et al., Safety Alignment Backfires: Preventing the Re-emergence of Suppressed Concepts in Fine-tuned Text-to-Image Diffusion Models, 2024.

**Additional Comments On Reviewer Discussion:**

There were initial concerns raised about the submission, some of which were adequately addressed during the rebuttal period, while others remained unresolved. During the reviewer discussion, the negative reviewers maintained their stance with supporting arguments, while the positive reviewers did not provide additional counterpoints.

---

### Decision · Program_Chairs · 2025-01-22

Accept (Poster)